# LEARNING DOCUMENT EMBEDDINGS WITH CNNS

## ABSTRACT

This paper proposes a new model for document embedding. Existing approaches either require complex inference or use recurrent neural networks that are difficult to parallelize. We take a different route and use recent advances in language modeling to develop a convolutional neural network embedding model. This allows us to train deeper architectures that are fully parallelizable. Stacking layers together increases the receptive filed allowing each successive layer to model increasingly longer range semantic dependences within the document. Empirically we demonstrate superior results on two publicly available benchmarks. Full code will be released with the final version of this paper.

## 1 INTRODUCTION

Document representation for machine reasoning remains a challenging open problem in natural language processing (NLP). A typical approach is to develop a document embedding model which produces fixed length vector representations that preserve relevant semantic information. These models are trained in unsupervised fashion on unlabeled text, and the resulting embeddings can be used as input for a variety of NLP tasks such as sentiment analysis and information retrieval (Blei et al., 2003; Le & Mikolov, 2014; Kiros et al., 2015). Despite significant research effort in this area the most commonly used methods are still based on the bag-of-words (n-grams) representations.

However, recent work has shown that remarkably accurate embedding models can be learned using distributed representations of words (Mikolov et al., 2013). Within this category two popular approaches are doc2vec (Le & Mikolov, 2014) and skip-thought (Kiros et al., 2015). doc2vec extends the distributed word model word2vec (Mikolov et al., 2013) by attaching document-specific vectors to word2vec and learning them jointly with word representations. While accurate, this model requires iterative optimization to be conducted for each new document making it challenging to deploy in high volume production environments. Furthermore, doc2vec is trained using localized contexts of very small size (typically 5 to 10 words) and never sees the whole document. This makes it difficult to capture long range semantic relationships within the document.

Skip-thought uses a recurrent neural network (RNN) to sequentially ingest the document one word at a time. Last layer activations after the last word are then taken as document embedding. RNN models have been gaining popularity and a number of other approaches have been proposed (Hill et al., 2016; Lin et al., 2017). Recurrent architecture addresses both problems of the doc2vec approach. During inference only a forward pass through the network is required to produce an embedding that is based on the entire content of the document. However, the sequential nature of the RNN makes it difficult to leverage the full benefits of modern hardware such as GPUs that offer highly scalable parallel execution. This can significantly slow down both training and inference. Consequently most RNN models including skip-thought are relatively shallow with only a few hidden layers. Moreover, many of the commonly used RNN achitectures such as LSTM (Hochreiter & Schmidhuber, 1997) and GRU (Chung et al., 2014), gate information form already seen input at each recurrence step. Repeated gating has an effect where more weight is placed on latter words and the network can "forget" earlier parts of the document (Lai et al., 2015). This is not ideal for document embedding where long range relationships that can occur anywhere in the document need to modeled.

In this work we propose an embedding model that addresses the aforementioned problems. We show that there is a direct connection between language and embedding models. We then use recent advances in language modeling to derive a convolutional neural network (CNN) embedding model. Similarly to skip-thought, inference in our model is done via a forward pass through the

CNN. However, the CNN architecture allows to process the entire document in parallel significantly accelerating both learning and inference. We show that the variable length input problem can be effectively dealt with using either padding or global pooling in the last convolutional layer. Moreover significant gains can be achieved using deeper architectures where each successive layer captures increasingly longer range dependencies in the document.

## 2 RELATED WORK

Early work on document embedding primarily focused on bag-of-words (BOW) and bag-of-ngrams representations (Harris, 1954). Despite significant effort in this area bag-of-words still remains one of the most popular and commonly used approaches. However, BOW models suffer from two major disadvantages, first, the dimensionality of BOW embeddings is proportional to the dictionary size often resulting in sparse and high dimensional embeddings. Second, BOW destroys the word order which in turn destroys much of the semantic structure. It is easy to find examples of documents that contain similar words but have very different meaning because of the word order. This makes it evident that BOW approach is limited in the type of semantic structure that it can represent. N-grams partially address this issues but quickly become impractical beyond 2-grams due to dimensionality explosion and rare sequence problems.

Many approaches have been proposed to improve the scalability and performance of BOW, including low-rank factorization models such as LSA (Deerwester et al., 1990) and pLSA (Hofmann, 1999), and topic models such as LDA (Blei et al., 2003). These models produce compact dense embeddings that typically outperform BOW. However, training is still done on the BOW view of each document which significantly limits the representational power.

Recently, distributed language approaches have become increasingly more popular. Here, words are represented using finite dimensional vectors that are concatenated together to form documents. Embedding models are then trained on these vectors sequences to produce compact fixed size representations that preserve semantic structure. This directly addresses the word order problem of the BOW models while avoiding dimensionality explosion. Approaches in this category include doc2vec (Le & Mikolov, 2014; Dai et al., 2015), skip-thought (Kiros et al., 2015) and others (Hill et al., 2016; Lin et al., 2017). Many of these models achieve state-of-the-art results with embeddings that are only a few hundred dimensions in length. However, they also have several disadvantages. For doc2vec, iterative optimization needs to be conducted during inference which makes it challenging to deploy this model in production. Skip-thought and other recently proposed RNN approaches avoid this problem, but virtually all currently used RNN architectures apply gating which places more weight on later words in the document. Moreover, RNNs are inherently sequential and are difficult to parallelize making them inefficient especially for long documents.

## 3 APPROACH

Given a document corpus $\mathbb{D} = \{D_1, \ldots, D_n\}$ the goal is to learn an embedding function $f$ that outputs a fixed length embedding for every $D \in \mathbb{D}$. The embedding needs to be compact and accurately summarize the semantic aspects of each document. It can then be used as input to NLP pipelines such as sentiment/topic classification and information retrieval. Note that besides documents in $\mathbb{D}$ we assume that no additional information is available and all training for $f$ is done in unsupervised fashion.

Each document $D$ contains a sequence of words $w_1, ..., w_{|D|}$ from a fixed dictionary $\mathcal{V}$. We use distributed representation and map every word in $\mathcal{V}$ to an $m$-dimensional vector. $\phi(w)$ denotes the vector representation for word $w$ which can be learned together with $f$ or initialized with a distributed word model such as word2vec (Mikolov et al., 2013). Concatenating together all word representations the input to the model becomes:

$$\phi(D) = [\phi(w_1), ..., \phi(w_{|D|})] \tag{1}$$

where $\phi(D)$ is an $m \times |D|$ matrix. The embedding function maps $\phi(D)$ to a fixed length vector $f(\phi(D), \theta) = \nu$ and $\theta$ is the set of free parameters to be learned. We use $D_{i:j} = w_i, w_{i+1}, \ldots w_j$ to denote the subsequence of words in $D$ and $f(\phi(D_{i:j}), \theta) = \nu_{i:j}$ to denote the corresponding embedding.

### 3.1 LANGUAGE MODELING

Inspired by the recent results in language modeling we explore the similarities between the two areas to derive our model for $f$. In language modeling the aim is to learn a probability model over the documents, i.e., $P(D) = P(w_1, \ldots, w_{|D|})$. The probability is typically factored into a product of conditional probabilities using the chain rule:

$$P(w_1, \ldots, w_{|D|}) = \prod_{i=1}^{|D|} P(w_i \mid w_1, \ldots, w_{i-1}), \tag{2}$$

and the models are trained to predict the next word $w_i$ given the subsequence $w_1, \ldots, w_{i-1}$. Recently, distributed representations have also become increasingly more popular in language modeling and virtually all current state-of-the-art approache use input representation similar to Equation 1 (Merity et al., 2016; Jozefowicz et al., 2016; Dauphin et al., 2016). While seemingly different, there is a close relationship between language modeling and document embedding. In both frameworks, the models are typically trained to predict a portion of the word sequence given a context. We explore this relationship in detail in this work and show that by altering the structure of the language model we get an architecture that can be used for document embedding.

To this end two recent advances in language modeling form the basis of our work. First, until recently RNNs were typically used to model $P(w_i \mid w_1, \ldots, w_{i-1})$. RNN language models process the word sequence one word at a time and thus require $O(|D|)$ sequential operations to generate predictions for a given document $D$. As we discussed above, these models can't take full advantage of parallel processing and thus scale poorly especially for long sequences. However, recently Dauphin et al. (2016) proposed a CNN-based language model where multiple layers of convolutions are applied to $\phi(D)$ to output the target probability distribution. CNN models are fully parallelizable, and Dauphin et al. (2016) show that deep CNN architectures with up to 14 layers can produce higher accuracy than leading RNN models while being over 20x more efficient at inference time. These results together with other related work (Kim, 2014; Kalchbrenner et al., 2014; Lai et al., 2015) in this area indicate that CNN models are effective and efficient alternative to RNNs for language tasks.

Second, traditionally $P(w_i \mid w_1, \ldots, w_{i-1})$ is modeled with a softmax layer where a separate weight vector is learned for every word $w \in \mathcal{V}$. However, given that in the input each word is already represented by a vector $\phi(w)$ recent work by Press & Wolf (2016) and Inan et al. (2017) simplified this model by reusing the weights:

$$P(w_i \mid w_1, \ldots, w_{i-1}) = \frac{\exp(\phi(w_i)^T \nu_{1:i-1})}{\sum_{w \in \mathcal{V}} \exp((\phi(w)^T \nu_{1:i-1})} \tag{3}$$

where $\nu_{1:i-1}$ is the $m$-dimensional output (last layer's activations) produced by the model after seeing the subsequence $w_1, \ldots, w_{i-1}$. Note that here the word representations $\phi(w)$ are used both as input and as weights in the softmax layer, and the last hidden layer of the model is fixed to have $m$ hidden units. The model is trained to output a vector $\nu_{1:i-1}$ that is "similar" to the representation of the next word $\phi(w_i)$. Reusing the representations reduces the number of parameters by close to 30% while producing comparable or better performance on many language modeling benchmarks (Press & Wolf, 2016; Inan et al., 2017). This indicates that the model is able to generate predictions directly in the $\phi$ space, and learn both $\phi$ and other layer weights simultaneously.

In this work we combine these ideas and propose a CNN model for document embedding. In the following sections we outline our model architecture in detail and present both learning and inference procedures.

### 3.2 EMBEDDING MODEL

At the core of our approach is the notion that a "good" embedding for a word sequence $w_1, \ldots, w_{i-1}$ should be an accurate predictor of the words $w_i, w_{i+1}, \ldots$ that follow. This notion similar to language modeling, however instead of predicting only the next word we *simultaneously* predict multiple words forward. Expanding the prediction to multiple words makes the problem more difficult since the only way to achieve that is by "understanding" the preceding sequence. This in turn forces the model to capture long range semantic structure in the document. To achieve this we note that in the simplified softmax from Equation 3 $\nu_{1:i-1}$ can be thought of as the embedding prediction for the

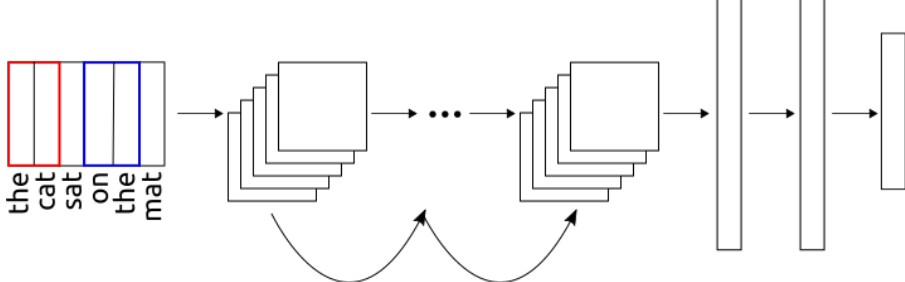

Figure 1: CNN embedding model diagram. Multiple layers of convolutions are applied to the distributed representation of the word sequence where each word is represented by an $m$ dimensional vector. The first convolutional layer contains kernels of size $m \times d$ that are applied to $d$ words at a time. Fully connected layers combine all activations from convolutions and map them to an $m$ dimensional embedding.

next word $w_i$. Furthermore, the probability of $w_i$ is proportional to $\exp(\phi(w_i)^T \nu_{1:i-1})$. Expanding this formulation to a window of $h$ words and framing the problem as binary classification we get the embedding loss function:

$$\mathcal{L}(i, D) = -\sum_{\substack{j=i \\ w_j \in D}}^{i+h} \log\left(\frac{1}{1 + \exp(-\phi(w_j)^T \nu_{1:i-1})}\right) - \sum_{w \notin D} \log\left(1 - \frac{1}{1 + \exp(-\phi(w)^T \nu_{1:i-1})}\right) \quad (4)$$

where $f(\phi(D_{1:i}), \theta) = \nu_{1:i}$ is the embedding for the subsequence $w_1, \ldots, w_i$. $\mathcal{L}$ aims to raise the probability of the $h$ words $w_i, w_{i+1} \ldots, w_{i+h}$ that immediately follow $w_{i-1}$ and lower it for all other words. This objective is similar to the negative sampling skip-gram model in word2vec/doc2vec (Mikolov et al., 2013). The main differences are that we only do forward prediction and $\nu_{1:i-1}$ is a function of all words up to $w_{i-1}$. In contrast, doc2vec incorporates backward prediction and uses the same $\nu$ for each $i$ in a given document that is updated directly. This significantly complicates inference for new documents. In contrast, our model addresses only requires a forward pass through $f$ during inference.

We discussed that CNN models have recently been shown to perform well on forward word prediction tasks with distributed representations, and are considerably more efficient than RNNs. Inspired by these results we propose to use a CNN model for $f$. In our approach multiple levels of convolutions are applied to $\phi(D)$. All convolutions are computed from left to right, and the first layer is composed of $m \times d$ kernels that operate on $d$ words at a time. Stacking layers together allows upper layers to model long range dependencies with receptive fields that span large sections of the document. Similarly to Dauphin et al. (2016), we found gated linear units (GLUs) to be helpful for learning deeper models, and use activation functions of the form:

$$h^l(x) = (h(x)^{l-1} * W^l + b^l) \cdot \sigma(h(x)^{l-1} * V^l + c^l) \quad (5)$$

where $h^l(x)$ is the output of $l$'th layer, $W^l$, $V^l$, $b^l$, $c^l$ are convolution parameters and $\sigma$ is the sigmoid function. Linear component of the GLU ensures that the gradient doesn't vanish through the layers, and sigmoid gating selectively chooses which information should be passed to the next layer. Empirically we found learning with this activation function to converge quickly and produce good results.

Traditional CNN models are not designed for variable length input so we investigate two ways to address this problem. The first approach is to apply zero padding to convert the input into fixed length:

$$\phi^k(D_{1:i}) = [\underbrace{\mathbf{0}, \ldots, \mathbf{0}}_{\max(k-i, 0)}, \phi(w_1), \phi(w_2), \ldots, \phi(w_{\min(i, t)})] \quad (6)$$

where $k$ is the target length. Sequences shorter than $k$ are left padded with $0$ vectors and those longer than $k$ are truncated after $k$ words. Analogous approach has been used by other CNN models for NLP (Dauphin et al., 2016; Conneau et al., 2017). While conceptually simple and easy to implement this approach has a drawback. For imbalanced datasets where document length varies significantly it is difficult to select $k$. Small $k$ leads to long documents being significantly truncated while large $k$ results in wasted computation on short documents.

To address this problem we note that convolutional layers in CNN can be straightforwardly applied to variable length input without modification. The problem arises in fully connected layers that can only operate on fixed length activations. We circumvent this by applying an aggregating function before passing the activations to fully connected layers. Formally, given unpadded representation $\phi(D)$ the activations from the last convolutional layer $h^l$ is a matrix where rows corresponds to kernels and columns depend on the length of $\phi(D)$. Since the number of kernels is fixed we can convert this matrix to a fixed length output by applying an aggregating function such as mean or max along the columns. This operation corresponds to global mean/max pooling commonly used in computer vision, and produces an output that can be passed to fully connected layers. Applying this approach eliminates the need for fixed length input and document padding/truncation which saves computation and makes the model more flexible. An alternative to global pooling is to use attention layer (Bahdanau et al., 2015), and in particular self attention (Lin et al., 2017) where the rows of $\phi(D)$ are first passed through a softmax functions and then self gated. However, this is beyond the scope of this paper and we leave it for future work.

### 3.3 LEARNING AND INFERENCE

During training we learn $f$ by minimizing the loss in Equation 4. Empirically we found that rather than fixing prediction point $i$ for each document, better results can be obtained with stochastic sampling. Specifically given a forward prediction window $h$ we repeatedly alternate between the following steps: (1) sample document $D \in \mathbb{D}$ (2) sample prediction point $i \in [\delta, |D| - h]$ (3) use gradients from $\mathcal{L}(i, D)$ to update $f$. Here $\delta > 0$ is an offset parameter to ensure that the model has enough context to do forward prediction. To speed up learning and improve convergence we conduct these steps using document mini batches and averaging the gradients across the mini batch. Note that separate prediction point $i$ is sampled for every document in the mini batch to encourage generalization. Second term in $\mathcal{L}(i, D)$ requires computing a summation over all words in the vocabulary which is prohibitively expensive. We address this by using a stochastic approximation with negative word samples. In practice we found that using small samples of 50 randomly sampled words is sufficient to achieve good results on all datasets. This learning algorithm is simple and straightforward to implement. Only two parameters $h$ and $\delta$ need to be tuned, and unlike skip-thought no sentence tokenizer is required. During inference $f$ is kept fixed and we conduct forward passes to generate embeddings for new documents.

## 4 EXPERIMENTS

To validate the proposed architecture, we conducted extensive experiments on two publicly available datasets: IMDB (Maas et al., 2011) and Amazon Fine Food Reviews (McAuley & Leskovec, 2013). We implemented our model using the TensorFlow library (Abadi et al., 2016). All experiments were conducted on a server with 6-core Intel i7-6800K @ 3.40GHz CPU, Nvidia GeForce GTX 1080 Ti GPU, and 64GB of RAM.

We found that initializing word embeddings with vectors from word2vec resulted in faster learning and often produced better performance than random initialization. This is consistent with other work in this area (Kim, 2014). We use the pre-trained word2vec vectors taken from the word2vec project page [1], and thus fix the input height to $m = 300$ for all models. Another advantage of training with word2vec is that the large pretrained vocabulary of over a million words and phrases can be used to apply our model to documents that contain previously unseen words. This is especially useful when the training set is small and has limited vocabulary.

| Model | tokens / s |
|---|---|
| skip-thought-uni | 27,493 |
| skip-thought-bi | 14,374 |
| CNN-pad | **312,744** |
| CNN-pool | 277,932 |

Table 1: Inference speed in tokens per second.

For all experiments, we use a six layer CNN architecture for our model with four convolutional layers and two fully connected layers. For the convolutional layers, we use 600 kernels per layer, residual connections every other layer (He et al., 2016), GLU activations (Dauphin et al., 2016) and batch normalization (Ioffe & Szegedy, 2015). ReLU activations are used in fully connected layers. The code with the full model architecture will be released with the final draft of this paper and we

---

[1]https://code.google.com/archive/p/word2vec

thus omit going into further details here. To address the variable length input problem we experiment with both padding (CNN-pad) and global poling (CNN-pool) approaches proposed in Section 3.2. For global pooling we found that max pooling produces better results than average pooling and use max pooling in all experiments. Max pooling has another advantage where by tracing the indexes of the max values chosen for each row back through the network we can infer which parts of the sequence the model is focusing on.

Embeddings from all models including baselines are evaluated by training a shallow classifier using the labeled training instances and we report the test set classification accuracy. This procedure is similar to the one conducted by Le & Mikolov (2014), and evaluates whether the model is able to capture semantic information accurately enough to do NLP tasks such as sentiment classification.

Table 1 shows inference speed in tokens (words) per second for uni-directional and bi-directional skip-thought models as well as our model. These results were generated by doing inference with batch size 1 to remove the effects of across batch GPU parallelization. From the table we see that the CNN architecture is over 10x faster than uni-directional skip-thought and over 20x faster than the bi-directional version. Similar results were shown by (Dauphin et al., 2016), and clearly demonstrate the advantage of using CNN over RNN architectures.

## 4.1 IMDB

The IMDB dataset is one of the largest publicly available sentiment analysis datasets collected from the IMDB database. This dataset consists of 50,000 movie reviews that are split evenly into 25,000 training and 25,000 test sets. There are no more than 30 reviews per movie to prevent the model from learning movie specific review patterns. The target sentiment labels are binarized: review scores $\leq 4$ are treated as negative and scores $\geq 7$ are treated as positive. In addition to labeled reviews, the dataset also contains 50,000 unlabeled reviews that can be used for unsupervised training.

The average review in this dataset contains approximately 230 words and we experiment with input length $k \in [400, 500, 600]$ for CNN-pad (see Equation 6). These values roughly correspond to 90'th, 95'th and 97'th percentiles of word length distribution and thus cover a significant portion of the dataset. In our experiments, we found that setting $k = 400$ produced good results. Furthermore, we were able to match over 90% of words to word2vec vectors and opt to simply drop the unmatched words.

| Method | Accuracy |
|---|---|
| N-gram | 86.50% |
| RNN-LM | 86.60% |
| NBSVM 3gram | **91.87%** |
| Avg. word2vec | 86.25% |
| skip-thought-2400 | 82.57% |
| skip-thought-600 | 83.44% |
| doc2vec-600 | 88.73% |
| CNN-pad | 88.74% |
| CNN-pool | 88.44% |

Table 2: IMDB results. Top row: supervised classifiers; bottom row: embedding models + classifier.

For our experiments, the goal is to evaluate the proposed CNN model and its ability to produce compact representations that accurately capture semantic aspects of documents. We use both labeled and unlabeled training reviews to train our model by using the objective outlined in Section 3.2. Training is done with mini batch gradient descent, using batch size of 100 and Adam optimizer (Kingma & Ba, 2014) with learning rate of 0.0003. For each document in the mini batch we sample prediction point $i$ and a set of negative words to make forward-backward passes through the CNN. Using parameter sweeps, we found that predicting $h = 10$ words forward with offset of $\delta = 10$ and 50 negative words produced good results.

We compare our model to an extensive set of baselines. These include word2vec, skip-thought and doc2vec. For word2vec (Avg. word2vec) we simply average the representations for all words that appear in a given document. For skip-thought we use the pre-trained model (skip-thought-2400) available from the authors' page, that has been trained on a large book corpus (Kiros et al., 2015). This model outputs significantly larger embeddings of size 2400. We also train another skip-thought model (skip-thought-600) on the IMDB training data using the default hyper-parameters and fixing the embedding size to 300 to match our model. In both cases, we report the results from the combined model, which combines the output embeddings from both a unidirectional and bidirectional encoder, as this always yields better results. We further train a doc2vec model using both the distributed bag-of-words and distributed memory methods, setting the encoding dimension

| Method | 2-class | 5-class |
|---|---|---|
| Avg. word2vec | 81.49% | 46.74% |
| skip-thought-2400 | 86.04% | **52.84**% |
| skip-thought-600 | 85.74% | 43.51% |
| doc2vec-600 | 86.58% | 47.34% |
| CNN-pad | **86.81**% | 52.58% |
| CNN-pool | 86.01% | 51.01% |

Table 4: AFFR results for 2 and 5 class sentiment classification tasks.

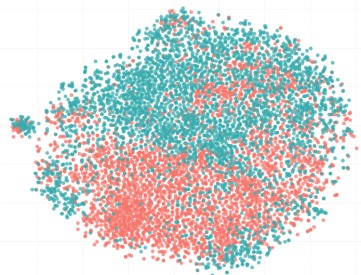

Figure 2: t-SNE representation of document embeddings produced by our model for the IMDB test set. Points are colored according to the sentiment label.

to 300 in both cases and using the hyper-parameters from Mesnil et al. (2014). We report results using an embedding that is a concatenation of both methods (doc2vec-600), as this gives the best result. Note that since both skip-thought and doc2vec baselines concatenate embeddings for best results, the final embeddings have dimension 600 and are twice the size of our model. Finally, all baselines are initialized with pre-trained word2vec.

Results from these experiments are shown in Table 2. For completeness, we also show results for various classification approaches trained in a fully supervised way, these are taken from Mesnil et al. (2014). From the table we see that our approach outperforms both word2vec and skip-thought baselines and performs comparable to doc2vec. Moreover, CNN-pool has comparable performance to CNN-pad suggesting that max pooling can be used to effectively get fixed length representation from variable length input. These results suggest that our model is able to learn a robust embedding function that accurately captures semantic information. Unlike doc2vec, once trained our model can be applied repeatedly to generated embeddings for new documents using a simple forward pass. This is a considerable advantage since inference in the CNN is fully deterministic and can be done in milliseconds on this dataset. It also worth nothing here that CNN language model proposed by Dauphin et al. (2016) used a much shorter context of less than 50 words to generate predictions. These results thus demonstrate that CNN models can also successfully model much longer sequences of over 2K words.

## 4.2 Amazon Fine Food Reviews

The Amazon Fine Food Reviews (AFFR) dataset is a collection of 568,454 reviews of Amazon food products left by users up to October 2012 (McAuley & Leskovec, 2013). Each example contains full text of the review, a short summary, and a rating of 1 to 5, which we use as the labels. This dataset does not come with a train-test split and is highly unbalanced. To address this, we perform our own split where we randomly sample 20,000 documents from each class and from these, we randomly select 16,000 for training and 4,000 for testing. This gives us training and test sets with 80,000 and 20,000 documents, respectively.

We train our model using the same architecture and training method as in IMDB experiments, with the only difference being that we set $k = 200$ for CNN-pad rather than $400$. This is due to shorter average length of the AFFR reviews compared to the IMDB reviews. Similarly to IMDB we compare against the same four baselines: word2vec, skip-thought-2400, skip-thought-600 and doc2vec-600. As before, we train a shallow sentiment classifier on top of the generated document embeddings but here we perform both binary 5-class classification. For binary classification, documents labeled 1 and 2 are treated as negative, 4 and 5 as positive, and we discard documents labeled 3. However, all training documents including those labeled 3 are used in the unsupervised phase.

Results for the classification task are shown in Table 3. Our approach performs comparably to the best baselines on each task. Highly competitive performance on the 5-way classification task indicates that our method is capable of successfully learning fine-grained differences between sentiment directly from unlabeled text. These results further support the conclusion that our proposed CNN architecture and learning framework produce robust embeddings that generalize well on NLP tasks of various complexity and size.

| | |
|---|---|
| **I found very little lobster in the can ... I also found I could purchase the same product at my local Publix market at less cost.** | |
| 0.755 | This is a decent can of herring although not my favorite ... I found the herring a little on the soft side but still enjoy them. |
| 0.738 | You'll love this if you plan to add seafood yourself to this pasta sauce ... Don't use this as your only pasta sauce. Too plain, boring ... |
| 0.733 | I read about the over abundance of lobster in Maine ... I am not paying 3 times the amount of the lobster tails for shipping |
| **This drink is horrible ... The coconut water tastes like some really watered down milk ... I would not recommend this to anyone.** | |
| 0.842 | I was really excited that coconut water came in flavors ... but it is way too strong and it tastes terrible ... save your money ... |
| 0.816 | wow, this stuff is bad. i drink all the brands, all the time ... It's awful ... I'm throwing the whole case away, no way to drink this. |
| 0.810 | This is the first coffee I tried when I got my Keurig. I was so disappointed in the flavor; tasted like plastic ... I would not recommend ... |
| **All I have to do is get the can out and my cat comes running.** | |
| 0.833 | Although expensive, these are really good. My cat can't wait to take his pills... |
| 0.787 | My kitty can't get enough of 'em. She loves them so much that she does anything to get them ... |
| 0.770 | Every time I open a can, my cat meows like CRAZY ... This is the only kind of food that I KNOW he likes. And it keeps him healthy. |

Table 5: Retrieval results on the AFFR dataset. For each query review shown in bold we retrieve top-3 most similar reviews using cosine distance between embeddings produced by our model. Cosine distance score is shown on the left for each retrieved result.

### 4.3 ANALYSIS

A common application of document embedding is information retrieval (Le & Mikolov, 2014) where the embedding vectors are indexed and used to quickly retrieve results for given a query. We use this approach to asses the quality of the embeddings that our model generates. Using the AFFR dataset we select several reviews as queries and retrieve top-3 most similar results using embedding cosine distance as similarity measure. The results are shown in Table 4.3, from this table we see that all retrieved reviews are highly relevant to each query both in content and in sentiment. The first group complains about seafood products, the second group is unhappy with a drink product and the last group are cat owners that all like a particular cat food product. Interestingly, the product in the retried reviews varies but both topic and sentiment stay consistent. For instance in the first group the three retrieved reviews are about herring, seafood pasta and lobster. However, similar to the query they are all negative and about seafood. This indicates that the model has learned the concepts of topic and sentiment without supervision and is able to successfully encode them into embeddings.

To get further visibility into the embeddings produced by our model we applied t-SNE to the embeddings inferred for the IMDB test set. t-SNE compresses the embedding vectors into two dimensions and we plot the corresponding two dimensional points coloring them according to the sentiment label. This plot is shown in Figure 2. From the figure we see a distinct separation between sentiment classes where most negative reviews are near the top and positive reviews are at the bottom. This further validates that the model is able to capture and encode sentiment information making the two classes near linearly separable.

### 5 CONCLUSION

We presented a CNN model for document embedding. In this approach successive layers of convolutions are applied to distributed word representations to model increasingly longer range semantic relationships within the document. We further proposed a stochastic forward prediction learning algorithm where the model is trained to predict the successive words for randomly chosen subsequences within the document. This learning procedure has few hyper parameters to tune and is straightforward to implement. Our model is able to take full advantage of parallel execution, and achieves better performance while also being significantly faster than current state-of-the-art RNN models.

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
