# OpenReview forum: "Learning Document Embeddings With CNNs"
_ICLR.cc/2018/Conference — Reject_

### Official Review · AnonReviewer2 · 2017-11-25
**Review of "Learning Document Embeddings With CNNs"**

**Rating:** 6
**Confidence:** 4

**Review:**

This paper proposes a new model for the general task of inducing document representations (embeddings). The approach uses a CNN architecture, distinguishing it from the majority of prior efforts on this problem, which have tended to use RNNs. This affords obvious computational advantages, as training may be parallelized.

Overall, the model presented is relatively simple (a good thing, in my view) and it indeed seems fast. I can thus see potential practical uses of this CNN based approach to document embedding in future work on language tasks. The training strategy, which entails selecting documents and then indexes within them stochastically, is also neat. Furthermore, the work is presented relatively clearly. That said, my main concerns regarding this paper are that: (1) there's not much new here, and, (2) the experimental setup may be flawed, in that it would seem model hyperparams were tuned for the proposed approach but not for the baselines; I elaborate on these concerns below.

Specific comments:
---
- It's hard to tease out exactly what's new here: the various elements used are all well known. But perhaps there is merit in putting the specific pieces together. Essentially, the novelty is using a CNN rather than an RNN to induce document embeddings.

- In Section 4.1, the authors write that they report results for their after running "parameter sweeps ..." -- I presume that these were performed on a validation set, but the authors should say so. In any case, a very potential weakness here: were analagous parameter sweeps for this dataset performed for the baseline models? It would seem not, as the authors write "the IMDB training data using the default hyper-parameters" for skip-thought. Surely it is unfair comparison if one model has been tuned to a given dataset while others use only the default hyper-parameters?

- Many important questions were left unaddressed in the experiments. For example, does one really need to use the gating mechanism borrowed from the Dauphin et al. paper? What happens if not? How big of an effect does the stochastic sampling of document indices have on the learned embeddings? Does the specific underlying CNN architecture affect results, and how much? None of these questions are explored.

- I was left a bit confused regarding how the v_{1:i-1} embedding is actually estimated; I think the details here are insufficient in the current presentation. The authors write that this is a "function of all words up to w_{i-1}". This would seem to imply that at test time, prediction is not in fact parallelizable, no? Yet this seems to be one of the main arguments the authors make in favor of the model (in contrast to RNN based methods). In fact, I think the authors are proposing using the (aggregated) filter activation vectors (h^l(x)) in eq. 5, but for some reason this is not made explicit.

Minor comments:

- In Eq. 4, should the product be element-wise to realize the desired gating (as per the Dauhpin paper)? This should be made explicit in the notation.

- On the bottom of page 3, the authors claim "Expanding the prediction to multiple words makes the problem more difficult since the only way to achieve that is by 'understanding' the preceding sequence." This claim should either by made more precise or removed. It is not clear exactly what is meant here, nor what evidence supports it.

- Commas are missing in a few. For example on page 2, probably want a comma after "in parallel" (before "significantly"); also after "parallelize" above "Approach".

- Page 4: "In contrast, our model addresses only requires" --> drop the "addresses".

---

> ### Author Response · Authors · 2017-12-18
> **Response**
>
> We would like to thank the reviewer for taking the time to review our work and for the insightful suggestions and comments. Below we address some of the main concerns that were brought up.
>
> Regarding the novelty, we believe that the novel aspect of our work is the end-to-end application of CNNs to document embedding. To the best of our knowledge CNNs have not been applied to unsupervised semantic learning before and most research has concentrated on RNNs. Our work demonstrates that with appropriate architecture and objective function CNNs can achieve comparable or better performance with 10x to 20x faster inference.
>
> All parameter sweeps were done on the validation set and the best model was then tested on the test set. We made an extensive effort to tune the baselines and fully acknowledge that fair comparison is very important. By “default hyper parameters” we meant settings such as number of layers, activation functions and optimizer as these are integral parts of each proposed model. All other parameters were extensively tuned for each baseline using the same parameter sweeps as in our model. Furthermore, doc2vec results are taken from Mensil et al and correspond to a highly tuned version of this baseline.
>
> We agree that further analysis of the proposed architecture would be informative and will included it in the revised draft. In short we observed the following: 1) gating activation function provided between 1% - 3% improvement over relu activations 2) stochastic sampling of prediction point for each document resulted in better generalization especially for datasets like IMDB where document lengths vary significantly 3) for CNN architecture we found that using more than 3 or 4 convolutional layers did not significantly improve performance and mostly resulted in slower training and inference runtimes.
>
> The embedding for the subsequence v_{1:i-1} is obtained by passing word sequence w_1,...,w_{i-1} through the CNN. To deal with the variable length problem we apply max (or max k) in the last convolutional layer which always ensures that the activation that are passed to the fully connected layers have the same length. The activations of the last fully connected layer are then taken as the embedding for v_{1:i-1}. At test time we pass the full word sequence w_1,...,w_|D| through the CNN to get the embedding for the entire document. Not that unlike RNN which would require |D| sequential operations, CNN can process the entire sequence in parallel thus significantly accelerating inference.

---

### Official Review · AnonReviewer3 · 2017-11-27
**An okay paper that fails to document its contribution**

**Rating:** 4
**Confidence:** 3

**Review:**

This paper uses CNNs to build document embeddings.  The main advantage over other methods is that CNNs are very fast.

First and foremost I think this: "The code with the full model architecture will be released … and we thus omit going into further details here."  is not acceptable.  Releasing code is commendable, but it is not a substitute for actually explaining what you have done.  This is especially true when the main contribution of the work is a network architecture.  If you're going to propose a specific architecture I expect you to actually tell me what it is.

I'm a bit confused by section 3.1 on language modelling.  I think the claim that it is showing "a direct connection to language modelling" and that "we explore this relationship in detail" are both very much overstated.  I think it would be more accurate to say this paper takes some tricks that people have used for language modelling and applies them to learning document embeddings.

This paper proposed both a model and a training objective, and I would have liked to see some attempt to disentangle their effect.  If there is indeed a direct connection between embedding models and language models then I would have also expected to see some feedback effect from document embedding to language modeling.  Does the embedding objective proposed here also lead to better language models?

Overall I do not see a substantial contribution from this paper. The main claims seem to be that CNNs are fast, and can be used for NLP, neither of which are new.

---

> ### Author Response · Authors · 2017-12-18
> **Response**
>
> We would like to thank the reviewer for taking the time to review our work and for the insightful suggestions and comments. Below we address some of the main concerns that were brought up.
>
> Regarding the novelty, we believe that we have proposed the first CNN model for document embedding. While CNNs have been recently used for language modelling we are not aware of any CNN model for document embedding. Empirically we have demonstrated that our approach can match or outperform RNN models that are traditionally used for this task with 10x to 20x improvement in inference speed. As such we believe that our approach is novel and further explores a promising direction of using CNNs in place of RNNs for NLP tasks.
>
> We understand that the connection to language modeling is unclear and will revise the draft accordingly. The main point that we are making is that the loss in Equation 4 reduces to language modelling loss if instead of h words forward we predict just one. So we are not just using some tricks, but rather show that the CNN language model of Dauphin et al can be generalized to document embedding by modifying the objective function and network architecture. While the embedding objective that we propose can be used to train a language model, we found that predicting more than one word forward does not improve language model accuracy and generally makes it worse. This is expected since language models always predict one word forward and our objective thus optimizes for a different task. We did however find that increasing the prediction window improves the quality of document embeddings since it forces the embedding model to model longer range semantic dependencies.
>
> Finally, we believe that we have provided sufficient details on model architecture including number of layers, layer size, activation function and optimization parameters (see Section 4). The details that were omitted are not critical for model understanding or reproducibility and given space constraints we opted to include further empirical results instead.

---

> > ### Comment · AnonReviewer3 · 2017-12-21
> > **Response to Response**
> >
> > I am not convinced the application of CNNs to document modeling alone is an interesting novelty.  CNNs have been previously applied to NLP in various ways, and the speed advantages have been noted, for example Neural Machine Translation in Linear Time https://arxiv.org/abs/1610.10099 and Attention is All You Need https://arxiv.org/abs/1706.03762 both note speed as an advantage.
> >
> > These works do not address documents; however,
> >
> > 1. Similar things have been done before, for example in http://www.datalab.uci.edu/papers/kdd2015_dimitris.pdf from 2015 which cites a 2014 paper for the architecture.
> >
> > 2. The extension to documents here is just treating documents as really long sentences, which is not very substantial.
> >
> > I would still like to see some attempt to disentangle the contribution of the model and the objective. If it is indeed the case that multi-step predictions make language models perform worse then why should I expect them to make embedding models better? I think this claimed connection should either be explored and exploited, or if it cannot be exploited then it should be dropped.
> >
> > The fact that this comment page already has more than one request for clarification on the model architecture suggests that sufficient details are not present in the paper.

---

### Official Review · AnonReviewer1 · 2017-11-28
**No comparison against recent SOTA in text representation**

**Rating:** 2
**Confidence:** 5

**Review:**

This paper proposes using CNNs with a skip-gram like objective as a fast way to output document embeddings and much faster compared to skip-thought and RNN type models.

While the problem is an important one, the paper only compares speed with the RNN-type model and doesn't make any inference speed comparison with paragraph vectors (the main competing baseline in the paper). Paragraph vectors are also parallelizable so it's not obvious that this method would be superior to it. The paper in the introduction also states that doc2vec is trained using localized contexts (5 to 10 words) and never sees the whole document. If this was the case then paragraph vectors wouldn't work when representing a whole document, which it already does as can be seen in table 2.

The paper also fails to compare with the significant amount of existing literature on state of the art document embeddings. Many of these are likely to be faster than the method described in the paper. For example:


Arora, S., Liang, Y., & Ma, T. A simple but tough-to-beat baseline for sentence embeddings. ICLR 2017.
Chen, M. Efficient vector representation for documents through corruption. ICLR 2017.

---

> ### Author Response · Authors · 2017-12-18
> **Response**
>
> We would like to thank the reviewer for taking the time to review our work and for the insightful suggestions and comments. Below we address some of the main concerns that were brought up.
>
> First, we do not compare with the speed of doc2vec since doc2vec requires optimization to be conducted during inference for each new document. This involves computing multiple gradient updates and applying them to the paragraph vector using an optimizer of choice. Regardless of the implementation, this procedure is an order of magnitude slower than making a single forward pass through an RNN/CNN. The doc2vec implementation that we have is at least 10x slower during inference than RNN. These findings are not new and have been discussed by authors of SkipThought and other related works. As such we do not believe that speed comparison with doc2vec is relevant here.
>
> Second, we’d like to thank the reviewer for pointing out the two related works and will add them in the next revision of our draft. However, both papers propose models that represent documents as (weighted) averages of word vectors. We do compare with word2vec average (“Avg. word2vec” baseline) although it is the equal weight version, and in addition have conducted further experiments to compare with these two models. Chen at al reports IMDB accuracy of 88.3% (Table 1 in that paper), and we got an accuracy of 87.4% using the code released by Arora at al. Neither of these beat our approach. Furthermore, while average word vectors would be computationally faster than CNN, the temporal order of the words is completely lost. One can create many examples of documents with very similar word counts but drastically different meaning due to the order in which these words appear. For “global” inference tasks such as sentiment classification, word order is not particularly important since even bag-of-words models produce strong performance. However, for more complex tasks such as q&a it becomes critical, and we believe that our approach provides a principled way to do unsupervised document learning that fully preserves temporal aspects while being significantly faster than RNNs.

---

### Public Comment · ~Marc_Jin1 · 2017-11-13
**Reveal the details of this paper**

Our team is currently considering reproducing your paper. However the details of this paper, which are vital for our reproduction, appear to be vague. For example, which "shallow classifier" do you use? Just wondering when you will reveal the details or the code.

---

> ### Author Response · Authors · 2017-11-14
> **Model details**
>
> Hi Marc,
>
> Thank you for taking the interest in our work. Below are some further details of our CNN model and the classifier, let us know if you have further questions. We are currently working on cleaning up and refactoring the code and aim to release it in the next few weeks.
>
> The classifier is a feed forward neural network with a single hidden layer and a tanh activation function. We train the classifier for 500 epochs, with a batch size of 100 and a momentum optimizer, with a learning rate of 0.0008 and momentum value of 0.9. We compute the test classification accuracy after every epoch and take the highest attained value for each model.
>
> For the CNN model we use dropout of 0.8 (prob to keep), 300-900 kernels in each convolutional layer, gating activation function and residual connections every other layer [see Dauphin et al ICML 2017 for analogous architecture]. Words are represented using a pre-trained word2vec model with 300 dimensions and we update word vectors together with CNN during training. We use mini-batches of size 100 and predict 10 words forward for each example in the mini-batch using 50 negative samples to balance the classification objective. All CNN models use Adam optimizer with a learning rate of 0.0003.

---

> > ### Public Comment · ~Eloise_Huang1 · 2017-11-28
> > **Detail of hidden layer in classifier**
> >
> > "The classifier is a feed forward neural network with a single hidden layer and a tanh activation function." What kind of hidden layer?

---

### Public Comment · ~Eloise_Huang1 · 2017-11-23
**When and where will the code be released?**

So I'm just wondering when and where will the code be released?

---

### Public Comment · (anonymous) · 2018-01-11
**Significantly worse than state of the art**

The reported accuracies for doc2vec on IMDB are wrong, presumably a consequence of a suboptimal re-implementation. In the doc2vec paper, they report accuracy of 92.58%, significantly higher than your reported doc2vec accuracy, 88.73%, and the accuracy for the proposed method, 90.15%. Given this extremely poor implementation of a baseline on IMDB, I also doubt the accuracy of the AFFR results, where you only beat doc2vec by less than a percent.

You should compare against this paper from openAI: https://arxiv.org/pdf/1704.01444.pdf

On IMDB, using a single neuron from their embedding they get 92.3%, significantly better than your 90.15%. Using all the neurons, they get 92.88%.

Given that the reported results are actually very poor relative to state of the art, and that the authors did not conduct a proper evaluation of their proposed method, I strongly recommend rejection.

---

### Decision · Program_Chairs · 2018-01-29
**ICLR 2018 Conference Acceptance Decision**

**Decision:**

Reject

**Comment:**

there are two separate ideas embedded in this submission; (1) language modelling (with the negative sampling objective by mikolov et al.) is a good objective to use for extracting document representation, and (2) CNN is a faster alternative to RNN's, both of which have been studied in similar contexts earlier (e.g., paragraph vectors, CNN classifiers and so on, most of which were pointed out by the reviewers already.) Unfortunately reading this manuscript does not reveal too clearly how these two ideas connect to each other (and are separate from each other) and are related to earlier approaches, which were again pointed out by the reviewers. in summary, i believe this manuscript requires more work to be accepted.